# Assessment of crop damage by rodent pests from experimental barley crop fields in Farta District, South Gondar, Ethiopia

**Bewketu Takele Wondifraw[1], Mesele Yihune Tamene[2]\*, Afework Bekele Simegn[2]**

**1** Department of Biology, Debre Markos University, Debre Markos, Ethiopia, **2** Department of Zoological Sciences, Addis Ababa University, Addis Ababa, Ethiopia

\* mesyih@gmail.com, mesele.yihune@aau.edu.et

**Data Availability Statement:** All relevant data are within the manuscript and its Supporting Information files.

**Funding:** The author(s) received no specific funding for this work.

## Abstract

This study was conducted in Farta district, south Gondar from 2019 to 2020 cropping years to identify rodent pest species and estimate damage caused on barley crops. Four independent barley crop fields (40 x 40 m each) were sampled randomly to estimate the loss. Two were located near Alemsaga Priority State Forest and the other two were away from the forest. Four (2 x 2 m) rodent exclusion plots were established at 10 m interval as control units in each selected experimental barley fields using fine wire mesh. Rodent pest species were collected using both Sherman and snap traps throughout the different crop growing stages. The damaged and undamaged barley tillers by pest rodents were counted on five 1 x 1 m randomly sampled quadrats for each selected experimental fields. Variations on pest rodent population between cropping years and sites were analyzed using Chi square test. The mean crop damages between cropping years and experimental field sites were analyzed using two way ANOVA. *Arvicanthis abyssinicus*, *Mastomys natalensis*, *Arvicanthis dembeensis*, *Mus musculus*, *Lophuromys simensis*, *Tachyoryctes splendens* and *Hystrix cristata* were identified as pest rodents in the study area. A total of 968 individual rodents (427 in 2019 and 541 in 2020) were trapped during the study period. There was a statistical variation ($\chi^2 = 13.42$, df = 1 and P<0.05) between trapped individuals of the two successive years. The crop fields near the forest were more vulnerable than away from the forest during both cropping years. Statistical variations was observed on mean crop losses between cropping years and experimental barley crop sites. The highest crop damage was seen at maturity stage and the lowest during sowing in all experimental plots and cropping years. The percentage of barley yield loss due to rodent pests was 21.7 kg ha$^{-1}$. The monetary value of this yield loss was equivalent to 4875 Birr (121.9 US\$ h$^{-1}$). Alemsaga Forest as shelter and conservation strategies like free of farmland from livestock and terracing for soil conservation have great role for the high rodent pest populations in the study area. Field sanitation, trapping and using restricted rodenticides like zinc phosphide are the possible recommendation to local farmers against rodent pests.

**Competing interests:** The authors have declared that no competing interests exist.

## Introduction

Agronomic pests are major issues in yield loss both in pre-harvest and post-harvest stages. Rodents are considered to be as one of the major pests of agricultural crops. Collectively, about 10% of the rodent species affect agriculture [1]. In Africa, 77 out of 395 rodent species are pests [2]. In Ethiopia, more than 6 out of 91 species of rodents identified as significant agricultural pests [3]. Species under the genus *Mastomys* and *Arvicanthis* are very common crop pests in Africa [4]. *Arvicantith abyssinicus* and *A. dembeensis* were common in maize fields of central Ethiopia [5]. Additionally, species such as *S. albipes*, *M. awashensis* and *A. dembeensis* are also the major pests in northern Ethiopia [6, 7]. In Africa, 50% pest rodents are reported by four countries; Tanzania (24.69%), Nigeria (8.64%), Ethiopia (8.64%), and Kenya (8.02%) [6].

Many findings showed that rodents are number one pest in agriculture, horticulture, forestry and public goods [8–10]. They damage human foods, and spoil by urine and droppings (reducing the sales value), gnawing electrical cables (cause fires) and transmitting diseases to humans [11]. Most of the damage in cereals and crops occur during the sensitive young seedling stage before harvest [12]. During outbreaks, rodents cause severe losses that place people at risk in their food security [6].

All over the world, rodents cause about 30% annual crop damage [13]. About 20% damage to maize plantation, 34 to 100% loss of young wheat and 34% loss of barley was reported in Western Kenya [12]. In Ethiopia, estimates indicate 15–40% loss on pulses and oil seed, 13–29% loss on root crops, 9–48% loss on coffee and 21–60% loss on cotton due to pest rodents [14]. [1] reported 26.4% damage on maize crops in central Ethiopia and 9–44% of cereal crop loss in northern Ethiopia [15]. [9] also reported 50% crop loss in Eastern Ethiopia. Rodents adversely affect rural communities by damaging agricultural crops and urbans by damaging all the properties [16] However, identification of rodent pest species in the study area is scarce. Moreover, experimental estimates of barley crop losses due to pest rodents are also limited. Therefore, the aim of this study was to identify the rodent pest species and status of pre-harvest damage on barley crops fields around Alemsaga Priority State Forest, south Gondar, Ethiopia.

## Materials and methods

### Study area description

The study was carried out near to the Alemsaga Priority State Forest in Farta district, south Gondar Zone of Amhara Regional State during 2019/20 successive cropping year. The study area is located 666 km away from Addis Ababa and about 100 km from Bahir Dar. The District lies between 11˚ 32' to 12˚ 03' latitude and 37˚ 31' to 38˚ 43' longitude with total area of 11788 ha. The altitude of the district ranges from 1920–4135 m asl. The annual temperature ranges from 9-25C˚ and rainfall of 900–1099 mm [17] (Fig 1).

### Field experiment set up

Agricultural field located at different sites, one near the forest (50–100 m) and the other away from the forest (over 500 m) were selected to conduct the experiment. From each site, two separated barley crop fields (40 x 40 m each) were sampled randomly for barley crop loss assessment. The minimum distance between each experimental plot was 200 m [11]. Four 2 x 2 m rodent exclusion plots were built as control unit in each experimental field. The rodent exclusion plots were constructed from fine wire mesh, 1.5 m high above the ground and fixed 50 cm deep underground [7, 11]. Each control plot was located at 10 m interval and represent different places in the crop fields. The barley seeds were sown after the early rainy season in June, reach milky stage at the end of July, mature at the middle of August and harvested in

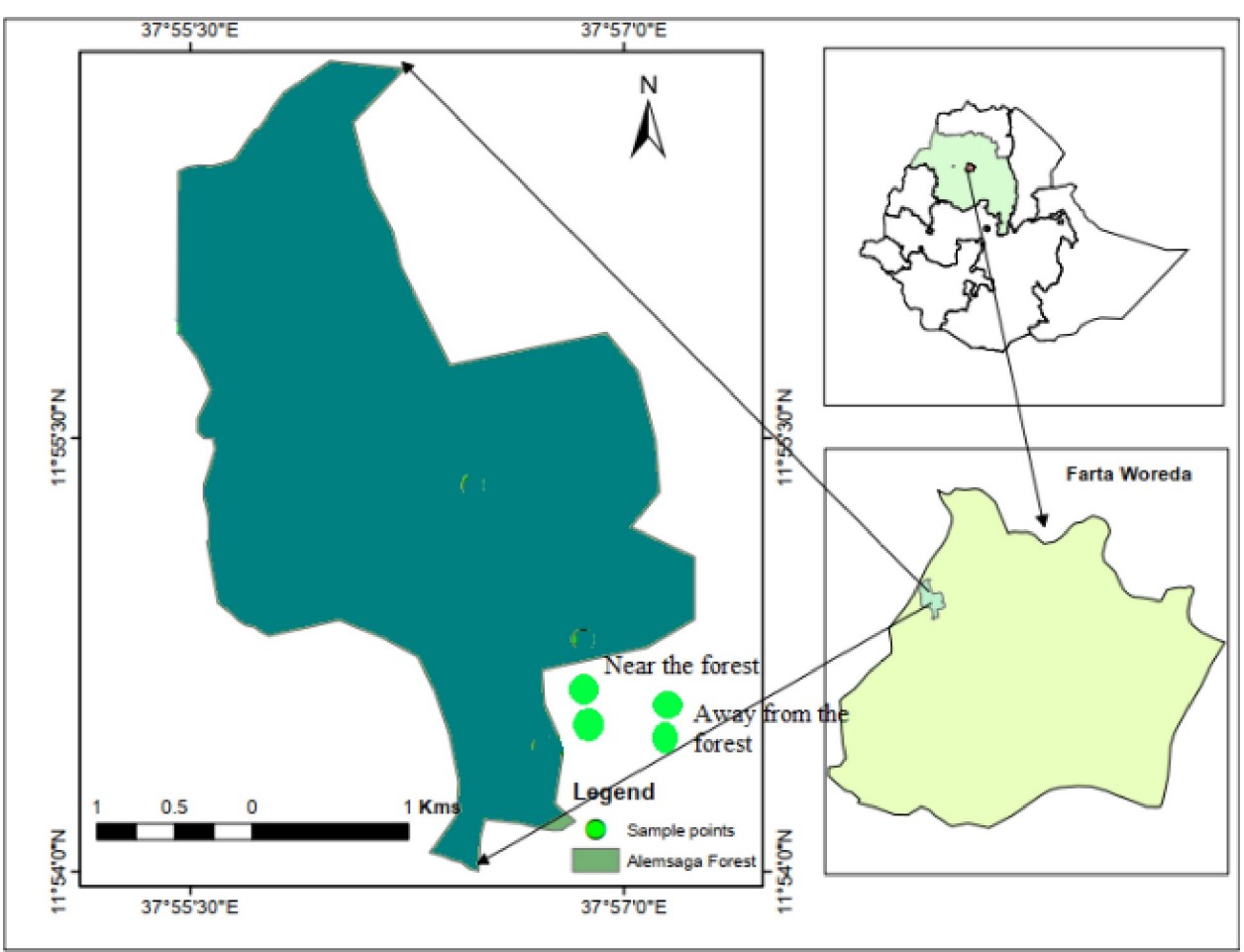

**Fig 1. Map of alemsaga priority state forest and sample areas (Arc GIS 10.4).**

September. During maturation, the exclusion plots were guarded to avoid birds. The crop type grown and agronomic practices were the same in all grids for two successive years. Permit for this work was obtained from south Gondar Zone Environmental Protection and Land Administration Bureau.

## Trapping of rodent pests

Rodent pest species were collected using both Sherman and snap traps from the different growing stages of the crop (during sowing, vegetative, booting and maturity stages before harvest) to determine the population dynamics of rodent pests. A total of 45 traps (25 live and 20 snap traps) were set within the crop field [7] for three consecutive days and nights at each growing stage of barley crop. The traps were set at every 10 m interval. The researcher and field assistants did regular follow-up to avoid damage and disturbance on experimental crop fields by other intruders such as birds and large mammals.

## Damage assessment

Damage assessment was carried out during the sowing, vegetative, booting and maturity stages before the farmers' intended date of harvest [7, 11]. Seed damage confirmation was carried out by

relating the actual plant rise counted in the sample quadrats with the possible rise from the control plots [7]. The number of damaged and undamaged tillers by pest rodents on the individual barley plants was counted for three days at every five days interval in each stage on randomly sampled five 1 x 1 m quadrats [7, 11]. High emphasis was given during sampling of quadrats (the middle and edges of the farmlands) (Fig 2). The characteristic oblique (45˚) cut on seedlings/tillers was used to confirm rodent damage [3, 11]. Finally, during harvest stage, barley yields in control plots (4 m$^2$) and equal proportion of the area from randomly selected experimental plots were taken and measured separately to observe yield variation as a result of rodent pests [11].

### Data analysis

Data were analyzed using SPSS software version 21. Chi square test was use to show pest rodent population variations between cropping years and selected barley crop sites. Seed losses during sowing were determined by comparing the actual plant emergence recorded in randomly selected sample quadrats with the potential emergence from the control plots. Crop damage that occurred during the vegetative, booting and maturity stages was estimated following [7] using the formula: % cut tillers = (a/b) 100 where, a = number of cut tillers in 1m by 1m sampled plots and b = total number of tillers in same sampled plots. The total barley yield loss was determined by comparing the yield in randomly sampled 2 by 2 m treatment plots of the same area control plots. The mean crop loss by pest rodents between cropping years and experimental barley crop sites was analyzed using two way ANOVA.

## Results

From two different barley fields, five rodent species were trapped and two species were observed as potential pests during the study period. These were *A. abyssinicus*, *M. natalensis*, *A. dembeensis*, *M. musculus*, *L. simensis*, *T. splendens* and *H. cristata*. *Tachyoryctes splendens* and *H. cristata* were observed as major pests in the area during the study period. Regardless of the sites, 968 individuals of different species (427 in 2019 and 541 in 2020) were recorded. There was a statistical significant variation ($\chi^2$ = 13.42, df = 1 and P<0.05) in the population of rodent pest species trapped in two successive years. Of the trapped pest rodents, *A. abyssinicus* and *M. natalensis* were the most prevalent species during both 2019 and 2020 cropping seasons. The populations of *A. dembeensis* showed increment from crop field near the forest to away from the forest. The population of rodent pests showed an increment from 2019 to 2020 except *A. dembeensis* that showed the reveres scenario.

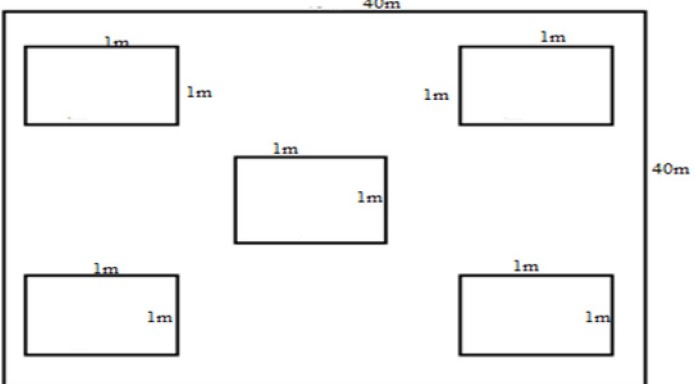

**Fig 2. A graphic representation of direct observation of crop loss assessment plots.**

The overall pest rodent abundance was high in the barley field away from the forest (518, 53.5%) than the barley field near the forest (450, 46.5%). There was a significant variation ($\chi^2$ = 4.78, df = 1 and p<0.05) on rodent pest populations between the two barley field sites (Table 1).

Trap success of most rodent pests during 2019 and 2020 cropping seasons was higher in the booting and maturation stages in both barley crop fields (Table 2). The highest trap success was recorded during booting stage of 2020 for *A. dembeensis* (40%) in the barley crop field away from the forest. However, zero trap success was obtained in sowing stage of both 2019 and 2020 cropping seasons. *Arvicanthis abyssinicus* and *M. musculus* were recorded in all stages of barley crops and cropping seasons.

Variations in rodent damage level were observed among the different growing stages. During both 2019 and 2020 cropping seasons, the highest damage was recorded at maturity stage (Table 3). The crop fields near the forest (CF1A, B) were more vulnerable than crop fields or plots selected away from the forest (CF2A, B) in both 2019 and 2020 cropping seasons. The peak crop damage (17.23%) was counted in the crop field near the forest (CF1B) during 2020 in the maturity stage. Sowing stage is the least in rodent damage in all experimental plots in both cropping years and cropping sites.

The experimental barley yield variations between treatment and control units of different cropping sites, plots as well as cropping years are represented in Table 4. Barley yield variations between the treatment and control units in each experimental plots were taken as loss due to pest rodents during the study period. The highest average crop loss was 0.18 (18%) recorded in 2020 cropping season on experimental barley field adjacent to the forest (CF1A, B). However, the least average crop damage was 0.09 (9%) recorded in 2019 cropping year from farmland away from the forest (CF2A, B). Statistical significant variation was observed on mean barley crop loss between cropping years and experimental crop field sites (P<0.05)The trends of average barley crop losses by pest rodents showed increment from 2019 to 2020 cropping years.

Based on the results obtained in experimental plots, the overall average barley yield in the control plot (4 m$^2$) was recorded to be 0.6 kg. Barley yield per hectare in the study area was estimated to be about 1500 kg or 15 quintal. The average loss because of rodent pests in treatment plots (4 m$^2$) was recorded to be 0.13 and the loss per hectare was estimated as 325 kg (3.25 quintal). Therefore, the percentage loss of the barley yield per hectare due to rodent pests (325/1500) was 21.7 kg ha$^{-1}$. Taking into consideration the price of barley crop and currency exchange rate for 2021 (~1500 Birr q-1 and 1 US$ = ~40 Birr, respectively), the monetary value for the average barley yield loss due to rodent pests was equivalent to 4875 Birr or121.9 US$ h$^{-1}$.

**Table 1. Pest rodent abundance in barley fields (2019 and 2020 cropping seasons).**

| Species | Total captured rodents | | | | | | | |
|---|---|---|---|---|---|---|---|---|
| | Crop field near the forest | | | | Crop field away from the forest | | | |
| | 2019 | | 2020 | | 2019 | | 2020 | |
| | Tot.cap | Re.ab (%) | Tot.cap | Re.ab (%) | Tot.cap | Re.ab (%) | Tot.cap | Re.ab (%) |
| *A. abyssinicus* | 79 | 38.92 | 82 | 33.2 | 88 | 39.3 | 101 | 34.4 |
| *M. natalensis* | 47 | 23.15 | 69 | 27.93 | 32 | 14.4 | 59 | 20.1 |
| *A. dembeensis* | 42 | 20.69 | 19 | 7.7 | 54 | 24.1 | 81 | 27.6 |
| *M. musculus* | 19 | 9.35 | 29 | 11.74 | 45 | 20.1 | 32 | 10.9 |
| *L. simensis* | 16 | 7.88 | 48 | 19.43 | 5 | 2.2 | 21 | 7.1 |
| Total | 203 | 100 | 247 | 100 | 224 | 100 | 294 | 100 |

Tot.cap = Total captured; Re.ab = Relative abundance).

**Table 2. Relative abundance and trap success of rodent pests at different growth stages of barley crop during 2019 and 2020 cropping seasons (A.ab = *A. abyssinicus*, M.na = *M. natalensis*, A.de = *A. dembeensis*, M.mu = *M. musculus*, L.si = *L. simensis*, Tra.su = trap success, Re.ab = relative abundance, C.F1 = crop field near the forest and CF2 = crop field away from the forest).**

| Cropping years | Species | Re.ab % and Tra. suc in each Crop growth stages (the numbers in the parenthesis indicated trap success in each crop stages | | | | | | | |
| --- | --- | --- | --- | --- | --- | --- | --- | --- | --- |
| | | C.F1 | | | | C.F2 | | | |
| | | Sowing | Vegetative | Booting | Maturation | Sowing | Vegetative | Booting | Maturation |
| 2019 | A. ab | 91.7 (8.1) | 30.4 (10.4) | 33.7 (25.9) | 46.3 (14.1) | 16.7 (2.2) | 39.2 (14.8) | 36.5 (31.1) | 57.7 (17.0) |
| | M. na | 0 | 32.6 (11.1) | 19.2 (14.8 | 29.3 (8.9) | 0 | 17.6 (6.7) | 16.5 (14.1) | 10.0 (2.9) |
| | A. de | 0 | 8.7 (2.9) | 30.8 (23.7) | 14.6 (4.4) | 38.9 (5.2) | 19.6 (7.4) | 22.6 (19.3) | 27.5 (8.1) |
| | M. mu | 8.2 (0.7) | 19.6 (6.7) | 8.7 (6.7) | 0 | 44.4 (5.9 | 19.6 (7.4) | 21.7 (18.5) | 5.0 (1.5) |
| | L. si | 0 (0) | 8.7 (2.9) | 7.7 (5.9) | 9.8 (2.9) | 0 (0) | 3.9 (1.5) | 2.6 (2.2) | 0 |
| 2020 | A. ab | 26.3 (3.7) | 43.9 (21.5) | 34.3 (25.9) | 21.7 (9.6) | 20.0 (2.9) | 48.5 (23.7) | 31.6 (37.0) | 30.0 (11.1) |
| | M. na | 0 (0) | 18.2 (8.9) | 30.4 (23.0) | 43.3 (19.3) | 30.0 (4.4) | 16.7 (8.1) | 15.2 (17.7) | 36.0 (13.3) |
| | A. de | 0 (0) | 7.6 (3.7) | 8.8 (6.7) | 8.3 (3.7) | 35.0 (5.2) | 16.6 (6.6) | 34.1 (40.0) | 22.0 (8.1) |
| | M. mu | 47.4 (6.7) | 6.1 (2.9) | 9.8 (7.4) | 10.0 (4.4) | 15.0 (2.2) | 16.6 (6.6) | 9.5 (11.1) | 10.0 (3.7) |
| | L. si | 26.3 (3.7) | 24.2 (11.9) | 16.7 (12.6) | 16.7 (7.4) | 0.0 (0.0) | 7.5 (3.7) | 9.5 (11.1) | 2.0 (0.7) |

## Discussion

Reporting of rodent pests with considerable crop losses in different parts of Ethiopia is not a new phenomenon. [7, 15, 18] reported rodents as the number one pests in central and northern Ethiopa. In the present study, five pest rodent species (*A. abyssinicus*, *M. natalensis*, *A. dembeensis*, *M. musculus* and *L. simensis*) were trapped and two were observed (*T. spledens* and *H. cristata*) from the barley crop fields at different stages (sowing, vegetative, booting and maturation) in two successive cropping seasons. *Arvicanthis abyssinicus*, *M. natalensis*, and *A. dembeensis* were the most prevalent species in both seasons. [3] reported the genus *Mastomys* and *Arvicanthis* are the most public agricultural pests with extensive distribution in Ethiopia. Similarly, [5] in central Ethiopia, [19] in Alleltu, Ethiopia, [20] around Arbaminch Forest, Southern Ethiopia, [21] in Wonch Suger Industry and [9] in Dire Dawa, Eastern Ethiopia reported the above rodents as major crop pests. In the northern parts of the country, [7] in Tigray region reported also these rodent species as the most prevalent agricultural pests. The relative abundance of pest rodents showed spatial and temporal variations in the present study. Relatively more populations of pest rodents were recorded in 2020 cropping season. This could be associated with extended rainy season in 2020 compared to 2019, enhancing rodents reproduction in the study area. [22] reported that rodents population increases during rainy season as essential resources are easily accessible. In line with the present study, [7] reported inter-annual changes in rodent relative abundance from experimental wheat and

**Table 3. Pest damage on different stages of barley crop at the two cropping sites in 2019 and 2020 cropping years (CF1A, B = Crop field near the forest, CF2A, B = crop field away from the forest).**

| Cropping years | Types of damage | Estimated damage (%) | | | | | |
| --- | --- | --- | --- | --- | --- | --- | --- |
| | | CF1A | CF1B | Average | CF2A | CF2B | Average |
| 2019 | Seed consumption | 0.79 | 1.9 | 1.35 | 0.5 | 0.6 | 0.55 |
| | Cut tillers | 13.9 | 12.12 | 13.01 | 11.1 | 9.52 | 10.31 |
| | Removal at maturity | 14.3 | 13.5 | 13.9 | 13.01 | 11.9 | 12.45 |
| 2020 | Seed consumption | 1.92 | 0.92 | 1.42 | 0.32 | 0.54 | 0.86 |
| | Cut tillers | 14.23 | 16.9 | 15.6 | 12.8 | 13.04 | 12.9 |
| | Removal at maturity | 15.04 | 17.23 | 16.14 | 15.03 | 12.91 | 13.97 |

**Table 4. Estimated pre-harvest yield loss of barley crop (CF1A, B = crop fields near the forest, CF2A,B = crop fields away from the forest).**

| Years | Experimental fields | Barley yield (kg) | | |
|---|---|---|---|---|
| | | Control units (4m$^2$) | Treatment units (4m$^2$) | Variation |
| 2019 | CFlA | 0.52 | 0.39 | 0.13 |
| | CF1B | 0.53 | 0.41 | 0.11 |
| | Average | 0.53 | 0.4 | 0.12 |
| | CF2A | 0.63 | 0.54 | 0.09 |
| | CF2B | 0.62 | 0.54 | 0.09 |
| | Average | 0.63 | 0.53 | 0.09 |
| | Total average | 0.58 | 0.47 | 0.11 |
| 2020 | CF1A | 0.60 | 0.43 | 0.17 |
| | CF1B | 0.62 | 0.44 | 0.18 |
| | Average | 0.61 | 0.44 | 0.18 |
| | CF2A | 0.62 | 0.52 | 0.10 |
| | CF2B | 0.64 | 0.53 | 0.11 |
| | Average | 0.63 | 0.53 | 0.11 |
| | Total average | 0.62 | 0.48 | 0.14 |

barley fields in Northern Ethiopia. Climatic oscillations and ecological disparities enhance rodent outbreaks [5]. During the study period, the relative abundance of rodents was higher in farmlands away from the forest than near to it. This might be due to the availability of frequent stone bundles (traces) constructed in the farmlands for water and soil conservation purposes. Similar finding was reported in the crop fields of northern Ethiopia [7]. The newly implemented soil conservation strategies like free of farmlands from livestock after harvest might have its own role in the study area. As stated by [23] livestock can disturb and reduce rodent pest populations when positioned in the farmlands after harvest.

High population of rodents was also recorded during the vegetative, booting and maturation stages than sowing stages in both sites and cropping seasons. [24] revealed relationship between rodent population and crop development as vegetation cover plays a great role in sheltering rodents. [7] reported that when crops advance towards the generative phases so did the accessibility and values for food and shelter, which may have been favourable for rodents reproduction and growth.

During the present study, damage assessment was carried out during sowing, vegetative and maturation stages of barley crops. The crop damage increases from seed consumption to vegetative as well as maturation stages. The result agrees with previous studies by [2, 3, 7, 9, 25]. The reports agree that rodent damage increases with crop phenology. [26] also described extreme rodent damages during maturation in cereal crops in Tigray region. In Tanzania, populations of *Mastomys natalensis* have caused yield losses up to 48%, and during acute outbreaks, the destruction has reached 80–100% of sowing and seedling stages in maize crops [6]. [9] also revealed that rodent damages in agricultural field mostly arises during early seedling phase and just before harvest. The possible reason might be due to flavoursome of crops and nature of predilection by the pest rodents and hoarding of panicles increase during the maturation period [9, 15]. The hoarded barley crop panicles observed in the burrows and stone bundles during the study period were also good indications for this. In addition, the barley crop maturation in the study area corresponds with the commencement of the dry season in which new populations could be engaged into the population. The present finding disagrees with the report of [6]. in which the highest losses (46%) occurred during early cropping stages and median loss (15%) during mature stage. The nature of rodent pest species and the nearby

habitats, conservation status and growth stages of crops can also affect the form and degree of crop damage by pest rodents [9]. Generally, the inherent characteristics of the pest species, change in climatic factors and farming practice are the main factors influencing the occurrence and severity of rodent damage on crops [3, 7].

Globally, 30% crops damaged by pest rodents during both pre-harvest and post-harvest stages were reported by [27]. In this study, the field experiment results showed 21.5% ha$^{-1}$ average losses of barley crops annually. The result is analogous with previous reports in Ethiopia by [7, 9]. However, the monetary value of the damage was high compared to the previous reports in Ethiopian. This might be the current inflation of the barley crop market value and the devalued condition of Ethiopian birr compared to $US. [3] also testified 26.4% loss on maize crop from maize farms of Ziway, Ethiopia. However, [25] reported lower result (9.6% of maize damage) from Bir Farm Development. In northern Ethiopia, [15] reported 8.9–44% yield loss annually to cereal crops due to pest rodents. Abroad Ethiopia, 2–5% in Australian, 40% in Hawai (U.S), 12–20% in South America and 20.7% in Andhra Pradesh (India) losses were reported from Sugarcane farms [28]. [29] in some parts of South America reported crop damage by pest rodents ranging from 5–90% of the total yield. In Tanzania, [8] also reported 80% loss on maize crops.

Even though the number of pest rodent individuals was less in barley crops selected near the forest, the damage was higher than crop fields away from the forest. This might be associated with selected crop fields away from the forest were surrounded by other barley crops that can share the damage by rodents. [11] also stated that the area with plantation provides refugia for pest rodents. The overall barley crop loss was relatively higher in 2019 cropping year than the 2020. The reason behind could be the outbreak of pest rodents population in 2020 as extended rainy period (from May to October) was observed in the study area. [30] described that 34–100% loss of wheat and barley and 20% loss of maize throughout the 1951 and 1962 outbreaks of rats in agricultural areas in Kenya during the extended rainy season. Generally, rodents can cause a significant financial damage to farmers. associated with their general feeding styles, high reproductive rate and ubiquiteus nature [6].

## Conclusions

The existence of Alemsaga Priority State Forest as a shelter, the newly implemented soil conservation strategies like free of farmland from livestock after harvest and terracing might have great role for the rodent pest populations in the area. Field sanitation, trapping, and using restricted of serious rodenticides including zinc phosphate are the possible solution to local farmers to mitigate the problem. The Woreda Environment Protection office should work with the local people to prevent crop damage due to rodent pests through awareness creation and implementing the recommended solution by the present study.

## Acknowledgments

We are grateful to the local farmers of the study are for their genuine help to do the field experiment.

## Author Contributions

**Data curation:** Bewketu Takele Wondifraw.

**Writing – review & editing:** Mesele Yihune Tamene, Afework Bekele Simegn.

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
