## [Decision Letter · Decision Letter 0]

21 Jun 2021

PONE-D-21-08035

Assessment of crop damage by rodent pests from experimental barley crop fields in Farta District, South Gondar, Ethiopia

PLOS ONE

Dear Dr. Tamene,

Thank you for submitting your manuscript to PLOS ONE. After careful consideration, we feel that it has merit but does not fully meet PLOS ONE’s publication criteria as it currently stands. Therefore, we invite you to submit a revised version of the manuscript that addresses the points raised during the review process.

We were really in difficult to find proper reviewers for this manuscript and please recommend several reviewers after your revisions.

We look forward to receiving your revised manuscript.

Kind regards,

Bi-Song Yue, Ph.D

Academic Editor

PLOS ONE

Journal Requirements:

1. Please ensure that your manuscript meets PLOS ONE's style requirements, including those for file naming. The PLOS ONE style templates can be found athttps://journals.plos.org/plosone/s/file?id=wjVg/PLOSOne_formatting_sample_main_body.pdf and https://journals.plos.org/plosone/s/file?id=ba62/PLOSOne_formatting_sample_title_authors_affiliations.pdf

'We would like to thank Addis Ababa University for financial support.'

'The author(s) received no specific funding for this work.'

Reviewers' comments:

Reviewer's Responses to Questions

**Comments to the Author**

1. Is the manuscript technically sound, and do the data support the conclusions?

Reviewer #1: Yes

2. Has the statistical analysis been performed appropriately and rigorously? 

Reviewer #1: Yes

3. Have the authors made all data underlying the findings in their manuscript fully available?

Reviewer #1: Yes

4. Is the manuscript presented in an intelligible fashion and written in standard English?

Reviewer #1: Yes

5. Review Comments to the Author

Reviewer #1: This study selected four independent barley crop fields (40 x 40 m each) as random sample plot to identify rodent pest species and estimate damage caused on barley crops in Farta district, south Gondar from 2019 to 2020 cropping years. This study had important guiding significance for assessing agricultural losses caused by rodent pests. There were several questions that need to be answered by the authors:

1. Apart from rodents, will there be other species that have caused damage to the crops in this area? Such as birds, insects, etc.? If insects can also cause damage to crops, is the control area enclosed by barbed wire not rigorous?

2. The author compared the crop damage caused by rodents in 2019 and 2020, but in fact, climatic conditions may also cause damage to crop harvests.Has the author considered the impact of climatic conditions on crop harvests during the past two years of starvation? Also, is there a difference in climate in areas far away from forests and near forests?

3. The author selected 2 areas (close to the forest and far from the forest), a total of 4 plots were selected for research. The overall study area is very large,so I think the selected sample plot(40�40m) a little small, and the authors selected 2 sample plot for each area (close to the forest and far from the forest), is the number of the sample plot for each area a little less?

4. Figures in the manuscript are not numbered

6. PLOS authors have the option to publish the peer review history of their article (what does this mean?). If published, this will include your full peer review and any attached files.

Reviewer #1: No

---

## [Author Response · Author response to Decision Letter 0]

28 Jun 2021

Responses to reviewers

• Page 4, 2nd paragraph, last line- the sentence “Permit for this work was obtained from south Gondar Zone Environmental Protection and Land Administration Bureau” added.

• Acknowledgment –the sentence –“ We would like to thank Addis Ababa University for financial support” deleted 

• Please keep the sentence “‘the author(s) received no specific funding for this work.” As it is under Funding Statement

• Manuscript has been amended according to Plos One writing style requirement. 

Q1. Apart from rodents, will there be other species that have caused damage to the crops in this area? Such as birds, insects, etc.? If insects can also cause damage to crops, is the control area enclosed by barbed wire not rigorous?

Answer: Yes, animals other than rodents like birds and insects can damage the crops. Birds during sowing (before ploughing) can pick the seeds and cause damages during maturation stage. But the sample plots were kept from birds by the researcher and field assistant farmers as it is explained in the methodology part of the manuscript (see methodology part marked in red (page 5 para 1, last three lines)). Moreover, birds do not damage seedling and boosting stages of the barley crops. So the researchers have considered the effects of birds on the experimental barley crop field during sowing and maturation periods.

Insects can also take the barley seeds during sowing (before ploughing). This taking of seeds takes place in both treatment and control unites as the control unites were excluded from rodents by fine wire mesh after ploughing. So there is no any effect on the result. However, after ploughing, the damage is negligible as insects do not dig and pick the barley seeds before germination as rodents do. But during sowing stage (before ploughing) the damage by insects was for both control and treatment units because the plots were excluded by fine wire mesh after ploughing. In addition, to increase its reliability, the researchers took 5 samples by considering the different parts (edge and middle) of each selected 40 by 40 m sample farmlands.

Q2. The author compared the crop damage caused by rodents in 2019 and 2020, but in fact, climatic conditions may also cause damage to crop harvests. Has the author considered the impact of climatic conditions on crop harvests during the past two years of harvest? Also, is there a difference in climate in areas far away from forests and near forests?

Answer: Yes, the climate condition may not be the same for the two cropping years (2019 and 2020). But, the intention of the experiment was to see the amount of yield variations between 2 by 2 m control and treatment plots. So the environment may equally affect both the treatment and control units equally unlike rodents damage. That means, the comparison of the yield was not simply taken as the yield variation between the two cropping years of the same plots. In addition, the experiment was taken on the same place and crop for two cropping years. So the climate variation may not have effect on the overall result. In addition to this, The distance between farmland near the forest and away from the forest is less than 500 m, may not have climatic variation. The aim of comparison of crop damage by rodents near the forest and away from the forest was to show the effect of the forest on the surrounding farmland crops hence the forest serves as a shelter for rodents. We conclude this fact from our finding.

Q3. The author selected 2 areas (close to the forest and far from the forest), a total of 4 plots were selected for research. The overall study area is very large, so I think the selected sample plot (40�40m) a little small, and the authors selected 2 sample plot for each area (close to the forest and far from the forest), is the number of the sample plot for each area a little less?

Answer: Yes, the researchers selected two 40 X 40 m plots of lands near the forest and the other two far from the forest. For each 40 X 40 m plots of lands, we sampled four 2 X 2 m control plots, 8 near the forest and the other 8 away from the forest (16 in total) for the experiment. In addition, for direct observation (in seed, seedling boosting and maturation stage damage assessments), five (5) 1 X 1 m samples for each 40 by 40 m plots of lands (20 in total) were taken. This helped the researcher to manage all samples efficiently for two successive cropping years. There is also homogeneity of sample areas in terms of crop grown and topography of the land, in this regard the sample size is enough for the study. 

Q4. Figures in the manuscript are not numbered

Answer: There are two figures in the manuscript and indicated in the text as Fig 1 (page 4, paragraph 1, line 3 and Fig 2 page 5 2nd paragraph, line 7).

---

## [Decision Letter · Decision Letter 1]

15 Jul 2021

Assessment of crop damage by rodent pests from experimental barley crop fields in Farta District, South Gondar, Ethiopia

PONE-D-21-08035R1

Dear Dr. Tamene,

We’re pleased to inform you that your manuscript has been judged scientifically suitable for publication and will be formally accepted for publication once it meets all outstanding technical requirements.

Kind regards,

Bi-Song Yue, Ph.D

Academic Editor

PLOS ONE

Reviewers' comments:

Reviewer's Responses to Questions

**Comments to the Author**

1. If the authors have adequately addressed your comments raised in a previous round of review and you feel that this manuscript is now acceptable for publication, you may indicate that here to bypass the “Comments to the Author” section, enter your conflict of interest statement in the “Confidential to Editor” section, and submit your "Accept" recommendation.

Reviewer #1: All comments have been addressed

2. Is the manuscript technically sound, and do the data support the conclusions?

Reviewer #1: Yes

3. Has the statistical analysis been performed appropriately and rigorously? 

Reviewer #1: Yes

4. Have the authors made all data underlying the findings in their manuscript fully available?

Reviewer #1: Yes

5. Is the manuscript presented in an intelligible fashion and written in standard English?

Reviewer #1: Yes

6. Review Comments to the Author

Reviewer #1: The autheors responded to all of my coments completely. All the suggested points are cleared. Therfore, this manuscript is acceptable.

7. PLOS authors have the option to publish the peer review history of their article (what does this mean?). If published, this will include your full peer review and any attached files.

Reviewer #1: No

---

## [Editor Report · Acceptance letter]

26 Jul 2021

PONE-D-21-08035R1 

Assessment of crop damage by rodent pests from experimental barley crop fields in Farta District, South Gondar, Ethiopia 

Dear Dr. Tamene:

I'm pleased to inform you that your manuscript has been deemed suitable for publication in PLOS ONE. Congratulations! Your manuscript is now with our production department. 

Kind regards, 

on behalf of

Dr. Bi-Song Yue 

Academic Editor

PLOS ONE